# Research on influencer marketing strategies based on double-layer network game theory

Zeguo Qiu[1,2]*, Hao Han[3], Yunhao Chen[1], Tianyu Wang[1]

1 School of Computer and Information Engineering, Harbin University of Commerce, Harbin, China,
2 Heilongjiang Key Laboratory of E-Commerce and Information Processing, Harbin, China, 3 School of Management, Harbin University of Commerce, Harbin, China

* qiuzg@hrbcu.edu.cn

## Abstract

The breakthroughs in communication technologies, such as 5G, have significantly accelerated the popularity of high-traffic consumption entertainment activities, including short video live streaming and real-time broadcasting, making them one of the most prevalent social interaction methods today. It is the high activity level of such online engagements that has given rise to diversified online marketing business models, opening up new channels and opportunities for interactions between brands and consumers. This study focuses on the emerging "influencer marketing" strategy rooted in content marketing, employing double-layer network game theory to construct a dual-layer relationship network between "brands" and "influencers" and establish a game-theoretic mechanism between them. During the construction of the influencer network, a novel concept—tunable clustering of influencers' followers—is specifically introduced, followed by an analysis of how micro-level decision-making factors (from brands and influencers) and network structures influence the evolutionary mechanisms of macro-level cooperative emergence. This study focuses on the emerging "influencer marketing" strategy rooted in content marketing, employing double-layer network game theory to construct a dual-layer relationship network between "brands" and "influencers", establishing a game-theoretic mechanism between them and analyzing how micro-level decision-making factors (from brands and influencers) influence the evolutionary mechanisms of macro-level cooperative emergence. Specifically, during the construction of the influencer network, the network structural metric—tunable clustering—is integrated with the practical scenario of uneven follower distribution among influencers, thereby investigating the impact of influencer network clustering intensity on the system's evolutionary dynamics. The research findings reveal that:(1) Influencer marketing represents a win-win cooperative model. (2) Brands' decision-making outcomes are significantly affected by profit margins, additional costs, and commission rates. (3) Creative incentives and tunable clustering predominantly shape influencers' decision-making behaviors. (4) Product

**Data availability statement:** All relevant data are within the paper and its Supporting Information files.

**Funding:** This research was supported in part by the Philosophy and Social Planning Project of Heilongjiang Province under Grant Nos. 22GLB105 and 23GLA011. Sponsor Qiu Zeguo is the first author of the thesis. He formulated the main research direction and content of this study and participated in the revision of the manuscript.

**Competing interests:** The authors have declared that no competing interests exist.

lifecycles and platform extraction rate impact decisions of both parties, with brands exhibiting higher sensitivity to environmental changes. Followers' trust levels in influencers have minimal influence on either party's decisions. Finally, applying reasonable values derived from parameter experiments to the influencer marketing model in the cosmetics industry demonstrates that this approach effectively enhances mutual benefits and stabilizes the overall business environment.

## 1. Introduction

The proliferation of mobile internet technologies has spurred the growth of content creation platforms—such as short video hubs, live-streaming services, and interactive forums—where creators engage audiences by sharing curated content. Through consistent high-quality output and distinctive personal branding, these creators evolve into influencers, individuals whose expertise and authenticity enable them to wield significant influence within niche domains.

While the term "online influencer" lacks a standardized definition, it is distinguished from "internet celebrity" by four key attributes [1]: (1) Content-Centricity: Production of original material conveying specialized knowledge or emotional resonance. (2) Audience Scale: A follower base meeting platform-specific thresholds. (3) Vertical Expertise: Authority in a specific domain (e.g., fashion, technology). (4) Commercial Efficacy: Demonstrated ability to drive brand promotion.

As the number of influencers on various platforms has significantly increased, brands have recognized the potential commercial value behind this influencer group and have begun to engage in advertising collaborations, brand promotions, and live-streaming sales with influencers, thereby gradually giving rise to the phenomenon known as influencer marketing.

Influencer marketing is a new commercial model wherein brands promote their products or services through influencers, transforming the followers of these influencers into customers. The core of influencer marketing lies in converting the influence of influencers into commercial value. Once influencers have amassed a sufficient number of followers, they can generate economic benefits through advertising, endorsements, live-streaming sales, and paid content. Additionally, they can leverage their influence to promote a brand or product, enhancing public acceptance of the brand and thereby helping businesses expand their market share. The rise of influencer marketing is attributed to the rapid development of social media, which provides influencers with platforms to showcase their talents and charm, while also offering followers opportunities to interact with influencers. This interactivity increases followers' willingness to purchase the products or services recommended by influencers [2], thereby propelling the growth of influencer marketing. The business model of influencer marketing is illustrated in Fig 1. Critically, the sustainability of this ecosystem hinges on effective collaboration between brands and influencers. As illustrated in Fig 1, their partnership forms the linchpin of a closed-loop system encompassing content creation, followers engagement, and revenue generation. Understanding the

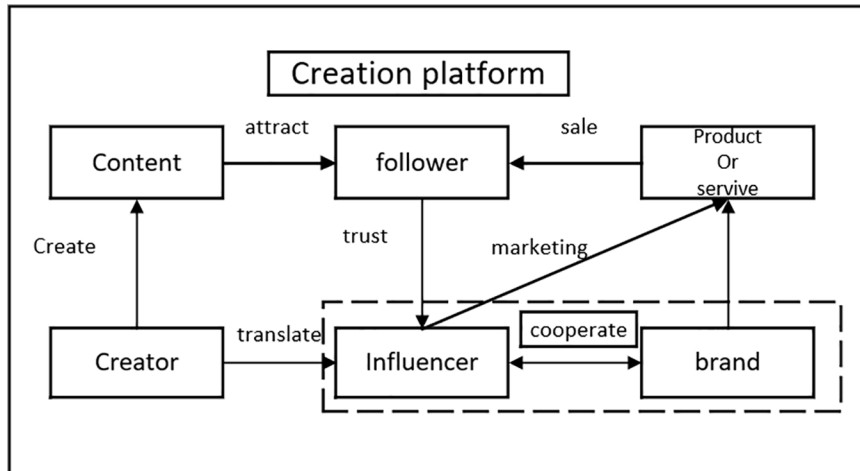

**Fig 1. Business Models for the Influencer Marketing.**

dynamics of this cooperation—particularly how micro-level decisions shape macro-level outcomes—is vital for optimizing benefit distribution and ensuring the model's long-term viability.

From the brand's perspective, brands establish connections with influencers and their followers. Compared to traditional advertising marketing, influencer marketing has characteristics such as shorter communication routes, efficient information dissemination, and stronger emotional engagement. For well-known brands, promoting through influencers can enhance brand recognition among followers compared to similar products, as well as improve brand reputation and purchase intention [3]. For brands with limited recognition, it can increase brand exposure and help identify potential customers within the influencer's follower group.

According to the two-step flow theory, individual influence is more frequent and effective than mass media in affecting people's decision-making. Xu Xiang's research indicates that the collectivization and similarity of opinion leaders' characteristics with those of their followers are high [4]. As opinion leaders for their followers, influencers share certain labels, characteristics, and thoughts with their fans. A group of followers will converge in a certain aspect, whether it be an interest, sympathy, or need. For example, followers of pet influencers all have a love for cute pets, followers of a scientific popularization influencer are all highly interested in the field the influencer covers, and followers of a postgraduate entrance exam influencer are mostly students or parents with exam needs. This prompts brands to shift their social media marketing strategies and seek collaborations with influencers. When collaborating with influencers, it is crucial to align with influencers whose labels match those of their own products. The high consistency between the brand and the influencer will generate positive attitudes towards the product, reducing its commercial perception [5], making it easier to successfully tap into potential customers online and improve conversion efficiency.

## 2. Literature review

### 2.1. Influencer marketing

Currently, research on influencer marketing in China is still in its infancy stage, with limited related studies available. Referring to industry reports such as the "2021 Douyin E-commerce Merchant Live Streaming White Paper," "2021 Douyin E-commerce Influencer Ecosystem Report," "2022 Douyin E-commerce Node Marketing White Paper","2022-2023 China Video KOL Marketing and Management White Paper," and relevant papers [6–15], these articles compares and analyzes

the characteristics of influencer marketing with emerging marketing models in recent years. The summary is presented in the Table 1 below.

Influencer marketing, as an emerging promotional paradigm, has garnered significant attention from academia and industry. Gurrieri et al. [16] delve into the complexities of influencer marketing through an interdisciplinary lens, examining how historical, economic, and technological shifts have shaped its evolution. Their work highlights the multifaceted identities of influencers and the strategies non-traditional influencers employ to connect with audiences. Crucially, they argue that influencer marketing has transcended its role as a mere marketing tactic to become a cultural phenomenon, one that both reflects and drives societal trends. Jiménez-Marín et al. [17] focus on the fashion industry, analyzing how influencers shape consumer brand choices. Grounded in Keller's Resonance Model, their mixed-methods study (combining qualitative and quantitative approaches) demonstrates that influencer-brand collaborations significantly enhance brand equity and consumer loyalty. This research provides empirical validation for integrating influencers into fashion branding strategies. Lee et al. [18] employ meta-analysis to compare the advertising efficacy of social media influencers (SMIs) and traditional celebrity endorsers. Synthesizing 39 experimental studies (2010–2024), they reveal that SMIs achieve comparable or superior results to celebrity endorsers and outperform brand-only campaigns. The study also identifies influencer credibility and tier as critical factors: mega-influencers exhibit stronger persuasive power, while nano-influencers lag in impact—a finding that informs strategic endorser selection. Sanz Marcos et al. [19] further explore how influencers amplify brand loyalty via the Resonance Model. Through in-depth interviews with Instagram influencers and large-scale follower surveys, they show that influencer-brand partnerships strengthen consumer-brand identification and foster long-term loyalty. This work offers empirical evidence for leveraging influencer collaborations to build enduring consumer relationships. Zozaya-Durazo et al. [20] examined influencer marketing strategies targeting minors aged 11–17, a sensitive consumer demographic. Their findings reveal that trust-based relationships between influencers and followers directly shape the perceived credibility of branded content. Notably, younger participants prioritized interactive engagement with influencers, while older adolescents emphasized content quality. These insights underscore the need for stringent ethical guidelines in

**Table 1. Comparison of Live E-Commerce, Internet Celebrity Economy and Influencer Marketing.**

| Marketing model | Live E-Commerce | Internet Celebrity Economy | Influencer Marketing |
|---|---|---|---|
| cooperator | anchor | Internet Celebrity | influencer |
| Whether to sign a contract with MCN | Yes | Yes | A small portion is signed (under contracts), while the majority operates independently. |
| Marketing approach | Live selling | hot news、IP Creation | content creation |
| Cash-out Methods | Live selling | virtual gifts、living | business order promotion、commission |
| Fan Stickiness | low | low | high |
| Evaluation index | GMV、Number of Fans | CTR、exposure | Number of Fans、Count of Original Content、view-through rate、ROI et al. |
| marketing features | 1.Relies on the personal charm of the anchor<br>2.Oligopoly<br>3. Node marketing<br>4. Competition between store broadcasting and official live streaming | 1. Relying on current events and trending topics;<br>2. Relying on the construction of Internet Celebrity images;<br>3. The quality of Internet Celebrity<br>3. The quality of internet celebrities varies greatly. | 1. rely on the continuous output of influencer-generated original content;<br>2. Beware of marketing accounts copying;<br>3. Personal reasons may affect whether an influencer chooses to monetize |

influencer marketing aimed at minors, urging brands to guard against excessive commercialization that may exert undue influence on youth values.

Currently, user growth across major internet platforms tends to stabilize, indicating that the user market and demand in the internet space are not expected to fluctuate significantly in the short term. One of the major challenges that companies face today is how to achieve revenue through internet marketing in an era where traffic is king. Another primary concern for brands is how to seize market opportunities among similar competitors. The influencer marketing offers an effective solution to these challenges.

## 2.2. Brand building

The strategies for brand building are diverse, including brand positioning [21], brand communication [22], and brand experience [23], among others. Brand positioning aims to clarify the core values of the brand and its target market. Brand communication disseminates brand information to consumers through means such as advertising, public relations, and social media. Brand experience emphasizes providing unique consumption experiences through products and services, thereby enhancing brand loyalty [24]. In recent years, methods such as content marketing [25], social media marketing [26], and experiential marketing [27] have been widely applied and have shown good results.

Companies should set objectives and performance measures for their marketing and other activities by understanding the process through which brand strategies and tactics provoke diverse, external reactions from corporate audiences, thereby creating brand value. Kumar and Srivastava (2020) discuss new perspectives on business model innovation in emerging markets [28], emphasizing the profound impact of business model innovation on brand building processes, and the emergence of influencer marketing as a brand marketing model innovation in line with the times. Maier and Wieringa (2021) provide insights into the impact of online market sales on retailers' own channel sales [29], albeit very minimal, but also emphasize the potential value of online markets in expanding retailers' customer base and enhancing brand awareness. The influencer marketing model also drives sales through influencers' online channels for brand owners' other channels to some extent. Gielens and Steenkamp (2019) explore the impact of digital (dis)intermediation on brand-building activities [30], particularly the disintermediation of digital marketing, which is at the core of the influencer marketing model.

Brand building is undergoing significant transformations against the backdrop of digitization and globalization. Online markets offer new growth opportunities for brands but also pose challenges to brand control. Emerging markets provide fertile ground for brand innovation, requiring brand strategies to adapt to local social, cultural, and economic environments. Digital decentralization phenomena necessitate that brands rethink their communication and transaction methods with consumers.

## 2.3. The application of double-layer network game theory

This study constructs a double-layer complex network evolutionary game model from the perspective of brand and influencer earnings, investigating the cooperation strategies of brands and influencers under the influencer marketing model. By simulating and quantifying the follower counts of influencers of the same type but at different levels on a platform using the Barabási-Albert (BA) growth model, this research explores the evolutionary trends of system cooperation under changes in environmental variables such as repurchase rate [31,32], product commission [33], platform extraction rate [34], and incentive rewards.

Complex network games integrate the theories of complex networks and game theory, abstracting game players as network nodes. By examining the intricate interactions among these nodes, this field aims to elucidate the strategic evolution of game players.

Presently, most applications of complex network games focus on single-layer network structures, involving games among homogeneous players [35–37]. In these studies, the players adopt the same pure strategies and can learn from the behaviors of any neighboring nodes to complete the game. Chen Hong constructed a complex network evolutionary

game model involving three players to investigate the branding of agricultural products in China [38]. As research progresses, the dynamic studies of complex networks have evolved into the exploration of multilayer networks [39]. Consequently, research on constructing multilayer network game models has begun to emerge [40]. This paper focuses on constructing a double-layer network model to explore multi-player game issues.

Network game theory offers distinct advantages over traditional evolutionary game methods. In classical evolutionary game theory, agents are boundedly rational, iteratively learning through trial-and-error processes to refine their strategies. The core focus lies in understanding how these agents learn, adapt strategies, and converge toward stable equilibria, with strategy shifts governed probabilistically by replicator dynamics that emulate "learning from experience." In contrast, double-layer network game theory abstracts each agent as a node, enabling strategy updates based on behavioral interactions between nodes. This framework allows for the design of specific strategy transition rules, a capability absent in traditional evolutionary approaches.

The unique strengths of network game theory make it particularly valuable for dissecting cooperative evolution in complex social systems, such as corporate competition and interpersonal relationships. Perc & Szolnoki [41], in their study of coevolutionary games, demonstrated that classical evolutionary game theory struggles to explain the persistence of cooperation under high defection pressures. Introducing co-evolutionary rules—such as synchronizing strategy updates with network topology, reputation mechanisms, or mobility—significantly enhances cooperation stability. This insight provides a theoretical foundation for dynamic corporate strategies, such as optimizing coordination networks or resource allocation rules to achieve co-evolved competitive advantages. Santos et al. [42] further revealed that the self-interested, autonomous rewiring of social networks by individuals is a core mechanism sustaining cooperation in highly connected heterogeneous networks. Applied to corporate competition, this finding suggests that organizations can overcome "social viscosity" constraints by dynamically forming alliances or optimizing supply chain networks (e.g., adjusting collaboration intensity and node connection weights) to drive cooperation emergence. In interpersonal relationship studies, the coevolution of strategies and social ties generates heterogeneous network structures, explaining the coexistence of cooperation and competition across multiple scales in real-world social networks. Additionally, multilayer network theory emphasizes the nonlinear effects of cross-layer coupling mechanisms—such as utility interdependencies, coordinated information flows, or strategy popularity propagation—on cooperative behaviors [43]. This framework aids in analyzing multilevel corporate ecosystems (e.g., cross-departmental collaboration or cross-industry alliances) and the coevolution of online-offline social networks. Their research shows that multilayer interactions amplify cooperative advantages through pattern formation and collective behavior, particularly under conditions of intense resource competition or information asymmetry.

In summary, network game theory systematically unravels the evolutionary pathways of cooperation in complex social systems by integrating co-evolutionary dynamics, adaptive network restructuring, and multilayer coupling mechanisms. This interdisciplinary framework lays the groundwork for optimizing corporate competitive strategies and managing interpersonal relationships in increasingly interconnected environments.

## 3. Construction of a double-layer network model for marketing strategies between brands and influencers in the influencer marketing

### 3.1. Construction of influencers network

Empirical studies reveal that degree distributions in many real-world networks exhibit power-law characteristics [44]. With advances in complex network theory, an increasing number of practical networks—including user relationship networks on the internet —have been identified as scale-free [45]. Given that both influencers and their followers are users on content creation platforms, their followership dynamics can be modeled using the Barabási-Albert (BA) network, which incorporates growth and preferential attachment mechanisms.

The attention network of all users on the creation platform is represented as $G1(V1,E1)$, where $V1$ denotes the set of all user nodes on the platform, and $E1$ represents the attention relationships among all users in the network. Assuming there are $n$ nodes in the network, the user network can be expressed using an adjacency matrix, as follows:

$$E1 = \begin{bmatrix} e1_{11} & e1_{12} & \cdots & e1_{1n} \\ e1_{21} & e1_{22} & \cdots & e1_{2n} \\ \vdots & \vdots & \ddots & \vdots \\ e1_{n1} & e1_{n1} & \cdots & e1_{nn} \end{bmatrix}$$

(1)

A node represents a user. If there is a relationship between two users $vi$ and $vj$ such that $e1i,j = 1$, then the directed edge from node $i$ to node $j$ represents that user $i$ follows user $j$, meaning $i$ is a fan of $j$. When the degree of a node reaches a certain level, that node can be considered an influencer node. Conversely, if $e1i,j = 0$, there is no follow behavior between the two users.

This study employs the Growing Scale-Free Networks with Tunable Clustering model proposed by Holme and Kim [46] to simulate user relationship networks on content platforms. By adjusting the selection probability $q$ during the edge-linking phase of network growth, we modulate the clustering coefficients of individual nodes and the entire network. A higher $q$ value corresponds to a larger global clustering coefficient, which structurally manifests as a higher proportion of followers concentrated around nodes with greater degrees (i.e., users with more followers). This parameter enables systematic exploration of how influencer follower scale—driven by network topology—shapes cooperative behaviors in subsequent experiments. To construct the influencer network, we retain the top-degree nodes from the simulated user network and record their follower counts.

### 3.2. Construction of brands network

In the information age, there is transparency among industries and enterprises. Basic business conditions of peer enterprises can be obtained through channels such as financial reports, stocks, and taxes. Additionally, data service platforms like some Data Analysis Platforms publish research reports on popular industries. Therefore, enterprises in various industries are in a state of perfect competition with equal competitive status [47]. Consequently, a fully connected network is used to construct the brand network, where all brands have the same competitive status in the influencer marketing. Let the brand relationship network be denoted as G2(V2, E2), where V2 represents all brand nodes and E2 indicates whether there is a competitive relationship among brands. The number of nodes in the brand network is determined by the number of influencer nodes, m, in the user network. The adjacency matrix representing the brand relationship network can be expressed as:

$$E2 = \begin{bmatrix} e2_{11} & e2_{12} & \cdots & e2_{1m} \\ e2_{21} & e2_{22} & \cdots & e2_{2m} \\ \vdots & \vdots & \ddots & \vdots \\ e2_{m1} & e2_{m1} & \cdots & e2_{mm} \end{bmatrix} = \begin{bmatrix} 0 & 1 & \cdots & 1 \\ 1 & 0 & \cdots & 1 \\ \vdots & \vdots & \ddots & \vdots \\ 1 & 1 & \cdots & 0 \end{bmatrix}$$

(2)

### 3.3. Construction of the connection between brand network and influencer network

In this paper, we construct a double-layer network model, with the brand network as the upper layer and the influencer network as the lower layer. Brands consider multiple factors when selecting influencers for cooperation, and often collaborate with multiple influencers for promotion. However, if an influencer has already established cooperation with other similar brands, the brand may be concerned about the influencer's sales efforts for their own products and the remaining

consumption potential of their fans. To ensure the independence of influencers connected to brands, this paper adopts a "one-to-one" connection between the two layers of the network, meaning that each influencer only connects with one brand for business purposes. Therefore, the link matrix E3 between the upper and lower networks is a diagonal matrix, which can be expressed as follows:

$$E3 = \begin{bmatrix} e3_{11} & 0 & \cdots & 0 \\ 0 & e3_{22} & \cdots & 0 \\ \vdots & \vdots & \ddots & \vdots \\ 0 & 0 & \cdots & e3_{mm} \end{bmatrix} = \begin{bmatrix} 1 & 0 & \cdots & 0 \\ 0 & 1 & \cdots & 0 \\ \vdots & \vdots & \ddots & \vdots \\ 0 & 0 & \cdots & 1 \end{bmatrix} \quad (3)$$

Fig 2 illustrates a simplified double-layer network model structure of "brands - influencers". In this model, the upper layer represents a fully connected network of brand nodes, while the lower layer represents the influencer network, which grows based on the BA scale-free network model. After ignoring non-influencer user nodes, the remaining influencer nodes form the influencer relationship network. The number of nodes in both the upper and lower layers is the same, and they are connected in a one-to-one relationship.

## 4. Construction of game model for marketing strategies between brands and influencers in the influencer marketing

### 4.1. Hypothesis of the game model

Brands and influencers function in a purely commercial collaboration model, where each party makes distinct behavior choices to secure corresponding gains. Within each group, there are typical social relationships present; individuals learn from one another and adapt their strategies to better suit the environment. This study is conducted from the perspective of brand owners, aiming to provide guidance for brands in selecting collaborating influencers, so it emphasizes brand interests in model assumptions.

**Hypothesis H1:** Brands have two pure strategies {conduct brand promotion, do not conduct brand promotion}. At the initial moment, each brand chooses a strategy according to probability $Pc$, where the proportion of brands implementing the brand promotion strategy is $x$, and the proportion of brands not conducting brand promotion is $1 - x$.

**Hypothesis H2:** Influencers have two pure strategies {cooperate, do not cooperate}. Similarly, they execute one of these strategies randomly according to probability $Pc$. The proportion of influencers cooperating with brands is $y$, and the proportion of influencers not cooperating with brands is $1 - y$.

**Hypothesis H3:** This study only considers online sales channels. When not conducting brand promotion, brands sell products through their own platforms or official stores on e-commerce platforms. When cooperating with influencers, sales are conducted through influencer channels (such as influencer homepage showcases, shopping carts under videos, and jump links). According to the principle of information asymmetry [48], it is assumed that the influencer's followers have not learned about the brand's product through other channels. Therefore, the followers' purchasing behavior

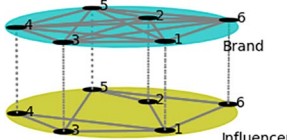

**Fig 2. Simple structure diagram of the double-layer network model of brands and influencers.**

represents a first-time consumption of the product. The consumption behavior of followers does not affect the sales volume of the regular online channels, and the two consumer groups are mutually independent and do not overlap. The product sales volumes for traditional online channels and influencer channels are denoted as $Q1$ and $Q2$, respectively. However, the promotion by influencers has a certain positive feedback effect on traditional online sales. In real-world scenarios, non-followers who are impressed by the influencers' promotion and subsequently purchase through search methods are defined as the synergy effect $a$ [49] generated when both parties cooperate. Let the cost per unit product be $C1$, the additional cost for promotion be $C2$, and the product price per unit be $P$.

**Hypothesis H4:** The benefits of influencers primarily come from three aspects. Firstly, there is the content creation incentive from the platform, which is a bonus pool established by content creation platforms to maintain their content ecosystem and encourage continuous output from creators. The remuneration is based on the data metrics of the content, such as views, likes, and shares. Since the calculation rules for this part of the benefits are determined by each platform and are greatly influenced by real-world factors, they are not easily unified and measured. This study quantifies the incentive benefits as a fixed value $L$. Secondly, there are the labor services for commercial orders, which are the primary source of income for influencers. This involves embedding brand products advertisements in their content creation or custom-making content for product promotion, thereby obtaining agreed-upon labor from the brand. According to current industry practices, the pricing for labor services $\lambda$ for waist and below influencers is generally 8% to 10% of their follower count, while head influencers usually have fixed prices for their labor services. Thirdly, there are product commissions, which are the commissions paid by brands to influencers when products listed in their homepage showcases are sold. The commission rate $\gamma$ varies significantly across industries, ranging from 30% to 70%. This study assumes that influencers only list the products of the collaborating brands in their showcases.

**Hypothesis H5:** Labor-service partnerships and commission-based collaborations operate as distinct models. The majority of influencer-brand partnerships function on a pure commission basis: influencers independently select products to sell, earning predetermined commission rates from brands without participating in formal promotional campaigns. In this model, product selection autonomy rests entirely with influencers. In contrast, labor-service partnerships follow a brand-driven process: brands proactively identify preferred influencers to negotiate marketing collaborations. Here, the initiative lies with brands, who dictate partnership terms and campaign objectives.

**Hypothesis H6:** This study does not consider the profit distribution between content creation platforms and e-commerce platforms, but rather focuses on the benefit conversion between brands and influencers. Some platforms have differences in functionalities or processes; for example, when purchasing products on Bilibili, users are redirected to external e-commerce platforms via official blue links. However, the core business model of influencer marketing remains unchanged, enabling both commission deductions and labor service cooperation. However, content creation platforms generally charge a 10% service fee for influencer commissions and labor income, with the deduction ratio being $R$.

**Hypothesis H7:** Based on the theory of consumer preferences, consumer preferences determine their purchasing behavior [50]. Influencers followed by fans possess certain characteristics and labels that align with consumer preferences. If a brand's products match the influencer's labels, the influencer's followers will be more receptive to the brand's products, and the conversion rate $\delta$ will increase. The conversion rate affects the cooperation behavior between both parties [51]. Research has shown that influencer promotion videos have a moderating effect on consumer purchasing behavior [52]. Considering multiple rounds of evolutionary game, the conversion rate after each iteration is defined as a logarithmic function, where $\delta = \log N_i$ (where $i = 1,2,3,\ldots,m$ $i = 1,2,3,\ldots,m$). At the same time, there may be a trust crisis among influencer followers towards the brand, which directly impacts purchasing behavior. Therefore, it is necessary to consider the trust level of followers towards the brand, represented by the parameter $\tau$ [53]. Parameter $\varepsilon$ represents the proportion of followers who have purchased products after the brand's collaboration with influencers.

**Hypothesis H8:** The evolutionary game model is a time sequence change model, requiring consideration of product wear and tear and lifespan $\rho$ during model iteration. Considering that consumers may have more than one consumption behavior throughout the evolution process, the repurchase rate $\theta$ is defined as the ratio of follower conversion rate to product lifespan. The higher the conversion rate, the greater the probability of followers repurchasing, and the shorter the product lifespan, the more frequent the repurchase behavior.

After completing the evolutionary game, the number and proportion of brands and collaborating influencers that still choose to promote are statistically analyzed to derive management insights.

## 4.2. Parameter definitions and profit matrix

The summary of the model parameters mentioned in the game theory model assumptions can be found in Table 2.

According to the model assumptions, the payoff matrix of the game process between the brand and the influencer can be established. The payoff situation for the brand and the influencer is as follows: The brand's revenue is divided into two parts: one is the sales revenue from other online channels (μ1), and the other is the sales revenue generated through influencer promotion (μ2). When promotions are conducted by influencers, the sales revenue from other online channels experiences a synergy effect. The influencer's revenue is composed of three parts: creative incentives (v1), sales commissions (v2), and contractual labor (v3). Among these, the actual earnings from v2 and v3 for the influencer need to be adjusted for platform extraction rate. For details, please refer to Table 3.

According to the assumptions, the specific values of each component of the brand's and the influencer's revenues are presented in formulas (4) to (10):

$$\mu_1 = Q_1 \left(P - C_1\right) R \tag{4}$$

$$\mu_2 = Q_2 \left(P(1 - \gamma) - C_1\right) \tag{5}$$

$$Q_2 = \varepsilon \theta N + (1 - \varepsilon)\delta N \tag{6}$$

$$\theta = \frac{\delta \tau}{\rho} \tag{7}$$

**Table 2. Description of the parameters of the game model.**

| Parameters related to the brand | Definition | Parameters related to the influencer | Definition |
|---|---|---|---|
| C1 | Cost per unit | $N_i$ | Number of influencer followers ($i = 1, 2\ldots\ldots, m$) |
| C2 | Promotion expenses and other costs. | p | Influencer fan niche level |
| Q1 | Sales volume from other online channels. | Q2 | Influencer channel sales volume |
| P | Unit price of the product | R | Platform extraction rate |
| ρ | Product life | L | Creator incentive |
| λ | Contract labor rate | δ | Follower conversion rate |
| α | Synergy benefit | θ | Repurchase rate |
| γ | Commission rate. | ε | Proportion of fans who have made a purchase |
| τ | Trust level of fans towards brands | q | Tunable clustering |

**Table 3. Revenue matrix of game among brand and influencer.**

| | | Influencer | |
| --- | --- | --- | --- |
| | | **Cooperate** | **Do not cooperate** |
| brand | promotion | $(1+\alpha)\mu_1+\mu_2-C_2-v_3$ <br> $(1-R)(v_2+v_3)$ | $\mu_1-C2$ <br> $v_1+(1-R)v_2$ |
| Do not promotion | $\mu_1$ | $\mu_1$ | |
| | $v_1$ | $v_1$ | |

$$v_1 = L \tag{8}$$

$$v_2 = Q_2 P \gamma \tag{9}$$

$$v_3 = \lambda N \tag{10}$$

Network game theory reflects the evolution trend of the group by altering individual behavior strategies. In this model, the decision vector of each individual is defined as $D_{i,k}$ ($i = 1, 2, 3, \ldots, m$; $k = 1, 2$). Here, i denotes the i-th pair of brand and influencer combinations, and $k = 1$ represents the strategy vector of the influencer individual node, while $k = 2$ represents the strategy vector of the brand individual node. During the game, the payoffs for a single influencer node and a single brand node are denoted as $U1_i$ and $U2_i$ respectively.

$$D_{i,k} = \begin{cases} (1,0); Individuals\,adopt\,cooperative\,strategies. \\ (0,1); Individuals\,adopt\,betrayal\,strategies. \end{cases} \tag{11}$$

$$U1_i = D_{i1} \begin{pmatrix} (1-R)(v_2+v_3) & v_1+(1-R)v_2 \\ v_1 & v_1 \end{pmatrix} D_{i2}^{\mathsf{T}} \tag{12}$$

$$U2_i = D_{i2} \begin{pmatrix} (1+\alpha)\mu_1+\mu_2-C_2-v_3 & \mu_1-C2 \\ \mu_1 & \mu_1 \end{pmatrix} D_{i1}^{\mathsf{T}} \tag{13}$$

## 4.3. Network evolutionary rule

After determining the network structure and game model, it is necessary to establish the strategy update rules for both brand parties and influencers to achieve a stable dynamic cooperative game process. After each round of the game, brand parties and influencers will compare their rewards with those of their same-layer network neighbors and decide with a certain probability whether to change their own strategies in the next round of the game.

This paper adopts the Fermi dynamical update rule for strategy adjustment [54], where the brand party (influencer) i randomly selects a neighbor brand party (influencer) j to compare the rewards obtained in this round. If the neighbor brand party (influencer) has a higher reward than its own, it will mimic the neighbor's strategy in the next round of the game with a certain probability. This imitation rule is cross-level gaming and same-layer comparison, with the learning probability given by equation (14):

$$W_{D_i \to D_j} = \frac{1}{1+e^{\left[\frac{U_i - U_j}{k}\right]}}$$

(14)

Here,$k(k>0)$ characterizes the noise effect, indicating that individuals are allowed to make irrational choices, meaning that individuals with lower rewards still have a small probability of having their strategies learned by individuals with higher rewards. In this paper, we refer to existing research [55] and fix the value of $k$ at 0.1 in the simulation experiments.

## 5. Numerical simulation analysis of the evolutionary game between brands and influencers in the influencer marketing

After establishing the brand party and influencer game network model and formulating the evolution rules, initial parameter values are set according to the revenue rules of various content creation platforms and industry conditions, and simulation analysis is conducted using MATLAB software. The specific initial assignment is shown in Table 4. The basis for assigning values is the self-media industry-related data reports collated and analyzed from research interviews and data platforms. It is stipulated that at the initial moment of the system, brand parties and influencers adopting the influencer promotion model each account for 20% of the total.

### 5.1. Impact of parameter adjustments related to brand parties on game strategies

At present, the promotion mode by influencers is more about the marketing choice of snacks, beauty, clothing, 3C products and other brands. According to the statistics of ' 2022–2023 China Video KOL Marketing and Management White Paper ', beauty makeup, 3C electronics and other products are highly dependent on social media for grass planting, and their delivery volume will increase by 47% and 86% year-on-year in 2022. The reason for this phenomenon is that there are many varieties of vertical goods in these categories, and there are endless niche brands in the market. Different consumers have different product experience for each brand. It is one of the more convenient ways for the brand to promote the product through the brand. At the same time, these categories of goods have the characteristics of low customer unit price, and high customer unit price goods rely more on the endorsement of head influencers. This is because consumers rely more on the reputation of official stores and influencers for high customer unit price goods. Cost-effective goods are more dependent on the tail. Therefore, when considering the impact of product pricing on the game strategy of both parties, the unit price of the product is limited to 100 or less to ensure that the product is within the consumption level of fans. According to Fig 3, the results show that the influencer marketing is a win-win cooperation model between the brands and the influencers. With the continuous growth of profits and without considering the interference of other external factors, both sides have rapidly evolved from the initial state of low cooperation intention to comprehensive cooperation.

Based on the product pricing P experiment, if the unit product cost C1 is increased at the same rate as P, the profit margin will gradually decrease under the same product profit. The calculation of the profit margin is shown in formula (15).

$$r = \frac{P - C1}{P} \times 100\%$$

(15)

This allows for the study of the sales performance of products with different profit margins under the influencer marketing model. According to the experimental results, brand parties can set the profit margins of their products based on actual

**Table 4. System Simulation Parameter Setting Table.**

| Parameter | x | y | C1 | C2 | Q1 | P | ρ | τ |
|---|---|---|---|---|---|---|---|---|
| initial value | 0.2 | 0.2 | 10 | 1000 | 100 | 20 | 1 | 1 |
| Parameter | λ | α | γ | R | L | δ | K | q |
| initial value | 8% | 0.1 | 10% | 0% | 100 | $\log^{Ni}$ | 0.1 | 0.7 |

circumstances. According to the results in Fig 4, brand parties will not engage in influencer promotion under low-profit scenarios; the marketing concept of "small profits but quick turnover" does not apply to influencer promotion.

Influencer promotion is a marketing approach that requires initial capital investment, and the expenditure of additional costs also holds significant importance in the influencer marketing. Since brand parties have a demand for influencer promotion, expenditures and roles centered around the influencer marketing are bound to emerge during its implementation.

Typically, when brand parties collaborate with influencers, they allocate a portion of their budget for promotion, known as the promotion fee. This promotion fee is used in various forms throughout the collaboration period, such as offering different types of "benefits" to influencer fans, coupons for purchasing products, additional gifts, etc.; or for promoting commercial content creation, covering transportation costs for inviting influencers to visit physical stores or factories, etc. Apart from promotion fees, the influencer marketing has also given rise to a new role within brand parties – the media role. The responsibilities of this role include, but are not limited to, brand publicity, planning, project follow-up, and budgeting for placements. In the influencer marketing, the media role primarily focuses on connecting brand parties with influencers, conveying both parties' needs, drafting collaboration plans, and serving to facilitate cooperation. Implementing influencer promotion inevitably requires hiring talents for the media role. All of the aforementioned are part of the additional cost C2 in the process of influencer promotion.

According to Fig 5, the influencer marketing is an investment-return negatively correlated economic model. Greater additional investment by brand parties does not necessarily lead to more benefits. This is because influencer promotion is a low-cost, efficient marketing method, usually requiring minimal investment to tap into most of the potential benefits. How brand parties can formulate appropriate promotion plans is a critical issue worthy of study in the influencer marketing model.

The product life directly affects consumers' repurchase rates; the longer the usage time, the more it impacts product sales [56]. Similarly, in the influencer marketing, the lifespan of a brand's products affects sales, which in turn influences the cooperation decisions between the brand and influencers. According to the experimental results in the Fig 6, brand parties are more sensitive to product lifespan than influencers. When the product lifespan exceeds 2, brand parties experience a shift in strategy due to reduced profits. On the other hand, the maximum acceptable product lifespan for

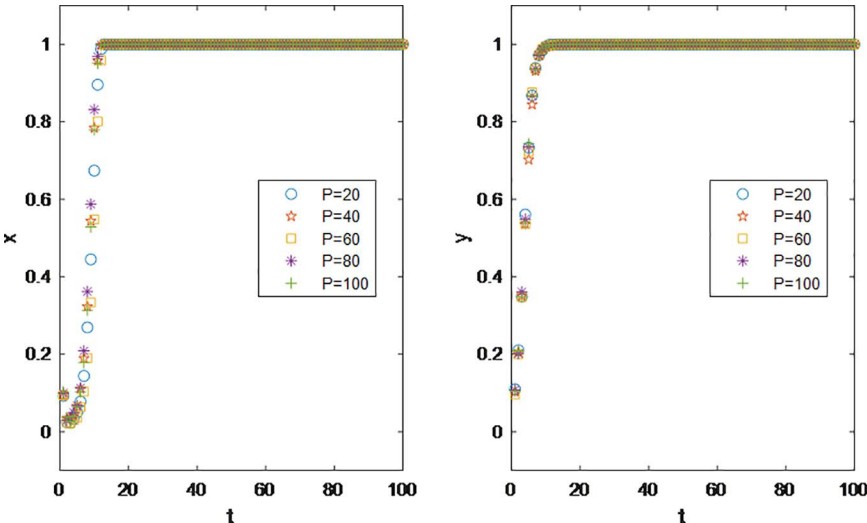

**Fig 3. Impact of Product Pricing P on Parties' Decision Making.**

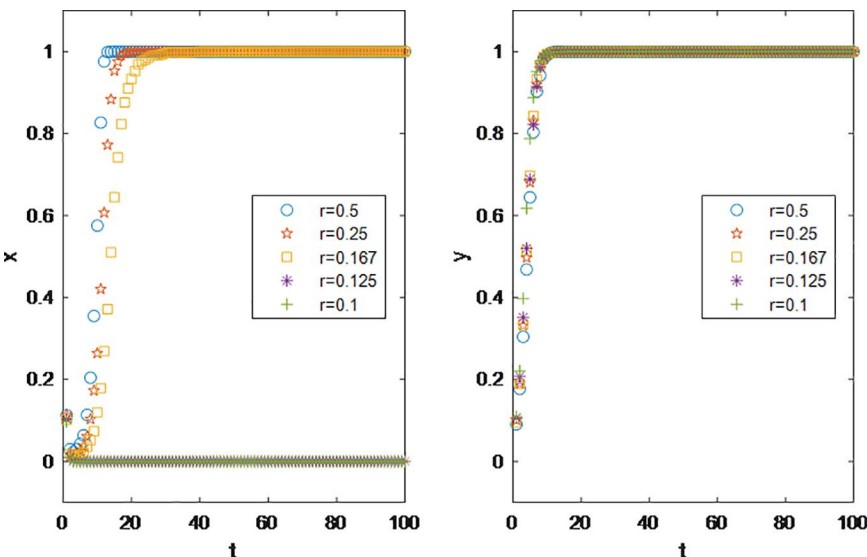

**Fig 4. Impact of Product cost *C1* on Parties' Decision Making.**

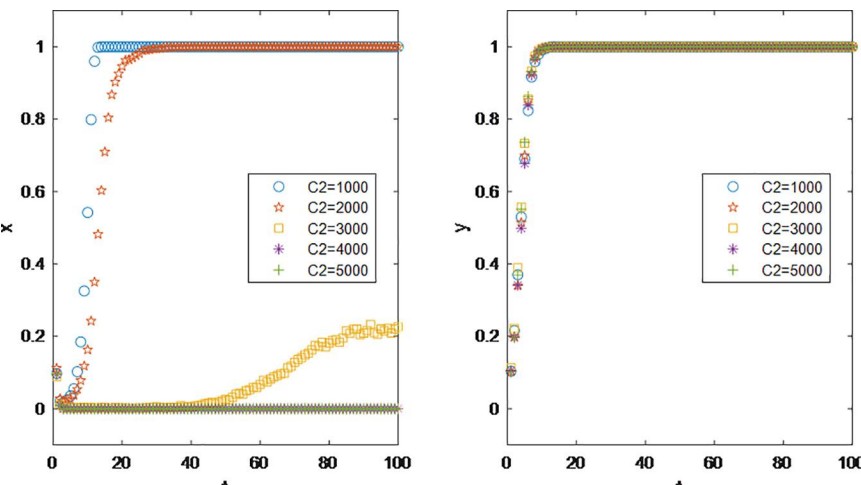

**Fig 5. Impact of Additional costs *C2* on Parties' Decision Making.**

influencers is 4. This indicates that in the influencer marketing, brand parties are more sensitive to changes in revenue, and their acceptable range for revenue fluctuations is narrower than that of influencers.

Commission in the realm of live-streaming e-commerce is mutually determined between brands and hosts on the basis of mutual acceptance through contractual agreements. However, in the influencer marketing, commissions are unilaterally set by brands when listing products on selection squares. The commission rate for the same product is uniform across all influencers, with the brands independently determining the rate. If influencers accept the commission rate offered by the brands, they can list the product in their showcases or 'little yellow carts' for sale; if not, they can choose not to list it.

Fig 7 illustrates that as the commission rate gradually increases, the profit margin of brands is compressed, leading to not only a decline in the stable proportion during cooperation but also a slowdown in the stabilization rate. According to

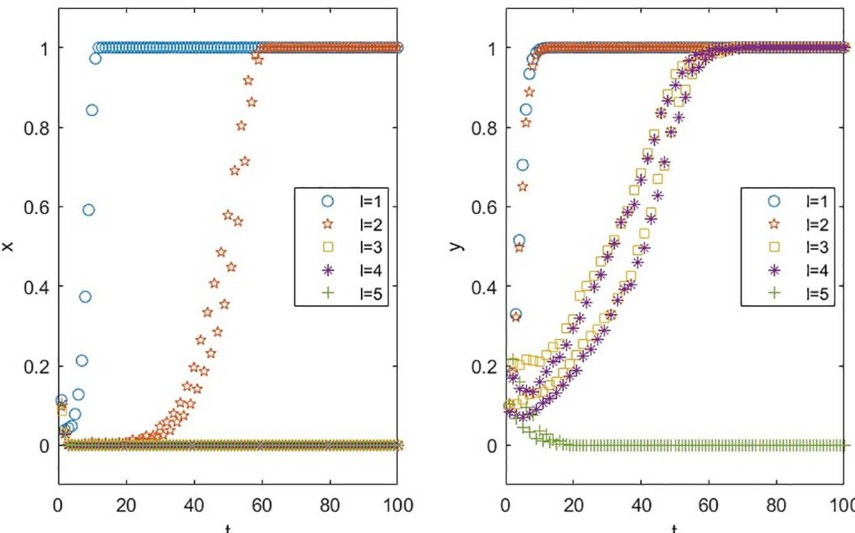

**Fig 6. Impact of Product life ρ on Parties' Decision Making.**

the initial values set, the profit margin for the low-priced products under study is 50%. When the commission rate is set at 40%, less than half of the brands still opt for cooperation, largely due to the large follower bases of the influencers they are connected with, ensuring that the return on investment remains positive. However, when the profit margin is fully converted into influencer commissions ($\gamma = 50\%$), at the evolutionary stability stage, no brands choose influencer marketing as a method of promotion.

### 5.2. Impact of parameter adjustments related to influencers on game strategies

The parameter Tunable Clustering ($q$) in scale-free networks with adjustable clustering quantifies the degree to which high-degree influencer nodes dominate follower distribution. Specifically, a higher $q$ value leads to a greater proportion of total followers concentrated among top influencers (i.e., nodes with the highest degrees), resulting in a polarized follower distribution where a few "head" influencers attract the majority of followers, while "long-tail" influencers retain minimal audiences. In real-world terms, this implies that influencers with larger follower bases wield greater influence within their vertical niches, as their ability to attract and retain followers reflects both specialized expertise and content appeal. Thus, q indirectly serves as a metric for quantifying an influencer's niche authority and professionalization level.

As illustrated in Fig 8, structural variations in the influencer network exhibit no discernible impact on brands' coordination decision-making. However, an increase in the network's tunable clustering parameter q correlates with a marked decline in influencers' willingness to partner with brands. This suggests that within the same vertical domain, polarized follower distributions—characterized by a dominance of followers concentrated among a few top influencers—create an environment less conducive to coordination engagements. Such polarization accelerates the attrition of long-tail influencers from the economic ecosystem, destabilizing the equilibrium of the influencer economy by disproportionately favoring "head" influencers. These insights advocate for policy interventions aimed at "de-centralizing" influencer economies, where redistributing economic benefits across all participants, rather than concentrating gains among top influencers, could mitigate systemic imbalances. By fostering inclusivity and equitable resource allocation, such strategies would enhance ecological resilience and drive sustainable macroeconomic growth.

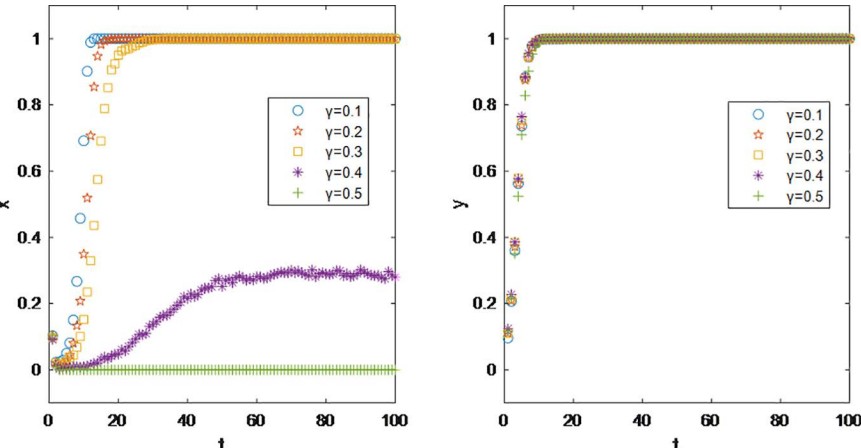

**Fig 7. Impact of Commission rate γ on Parties' Decision Making.**

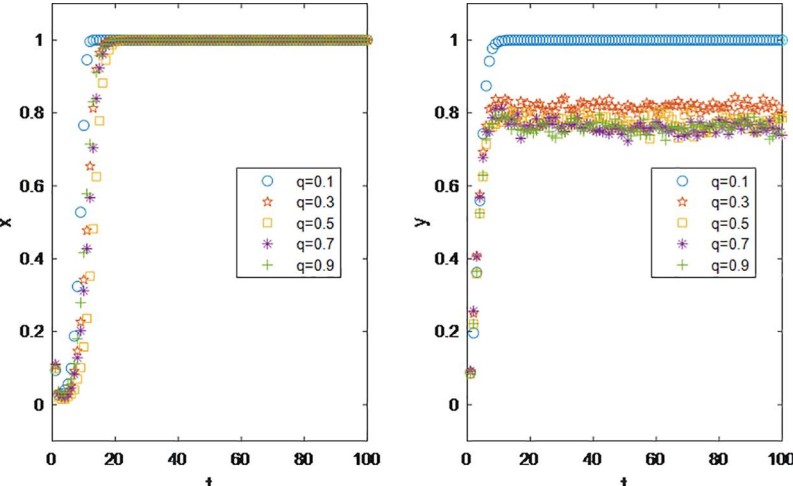

**Fig 8. Impact of Tunable Clustering q on Parties' Decision Making.**

The revenue sharing extracted by the creation platform in the influencer promotion model originates from the commission on products, if an influencer's income on the platform reaches the taxable level, the sharing portion typically includes the taxes that the influencer is liable to pay. According to the results in Fig 9, when the platform does not take a cut or the commission rate is relatively low, it does not affect the normal operation of influencer marketing. However, when the platform's defined commission rate is excessively high, some influencers may choose to abandon collaboration with brands, opting instead to focus on content creation and operating their accounts through incentive earnings. When the commission rate reaches 75%, only about half of the influencers choose to continue cooperating with brands.

For brands, although profits in traditional online sales channels are still subject to commission deductions by the platform, the influence of influencer marketing and the resulting sales far exceed those of traditional online sales. In addition, because the platform extraction is included in the commission. Therefore, brands are unaffected by the commission rate until the rate reaches 100%, at which point no influencers would likely choose to cooperate. To avoid incurring additional cost $C2$, they would shift to a noncooperation strategy.

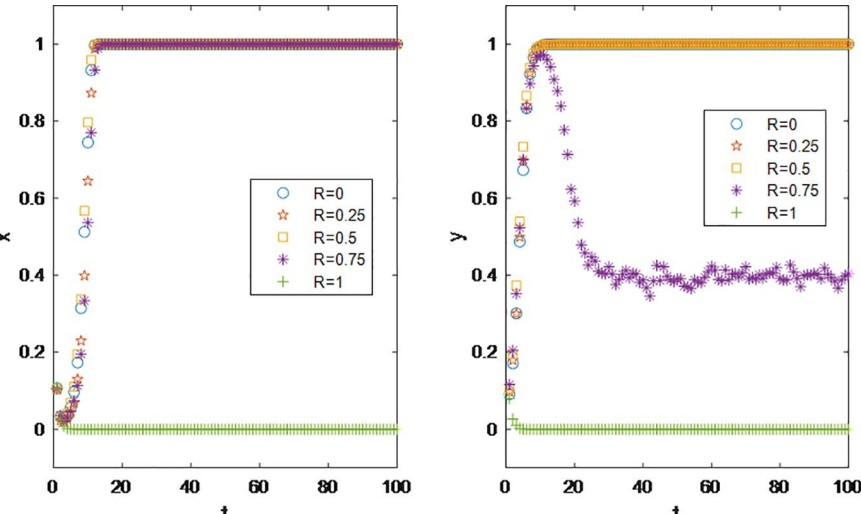

**Fig 9. Impact of platform extraction rate _R_ on Parties' Decision Making.**

Based on the rules of creation incentives across various platforms, the range for the parameter _L_ is set between 400 and 2000. In the current influencer marketing model, the majority of influencers engage in the "influencer" role as a side job or hobby, and for these content creators, income from brand contracts is not essential. Only a small portion of full-time influencers or those signed to MCN agencies are heavily reliant on business income from brands (including contract labor and commission on sales). Notably, commercial placements can affect the viewing experience of fans consuming the influencer's content, leading to degraded video metrics. Therefore, it is crucial to carefully consider the target audience, brand background, and potential impact on the audience when embedding commercial ads in videos [57]. As video traffic directly affects an influencer's earnings from participating in various incentive programs on the platform, some influencers choose not to engage in brand commerce to ensure the quality of their creations. These influencers may opt for a strategy of not collaborating with brands when their income from platform incentives and business revenue are balanced.

According to the results in Fig 10, in the initial stage, due to the unexplored purchasing power of their followers, an influencer's business income exceeds their incentive earnings, leading them to choose to collaborate with brands. However, in the later stage of the game, due to a decline in followers' purchasing power (_ε_ gradually approaching 1) and a decrease in repeat purchases, the incentive earnings of some creators gradually surpass their business income. Consequently, during system stabilization, the higher the incentive earnings, the lower the proportion of influencers choosing to cooperate with brands. When incentive earnings reach the maximum of 2000, only around 10% of influencers continue to cooperate. These predominantly include the top 10% of influencers in terms of follower count, whose income is no longer dependent on platform incentives. The above analysis indicates that influencer marketing is an economic model that heavily relies on the influencer's follower base and purchasing power.

The experimental results from Fig 11 show that the level of trust fans have in influencers plays a moderating role in the model, but does not truly affect the actual operation of the influencer marketing. No matter how the value of _τ_ is adjusted, the evolutionary trend of the model remains unchanged. Since _τ_ affects the brand's final revenue through its impact on product sales via the influencer channel, and given the large number of influencer fans, its ultimate impact on the system is diminished.

From the analysis of the influencer marketing model, it is clear that brand parties need to consider the overall profitability of their teams and organizations, while influencers participate in the promotion process as individuals with greater freedom.

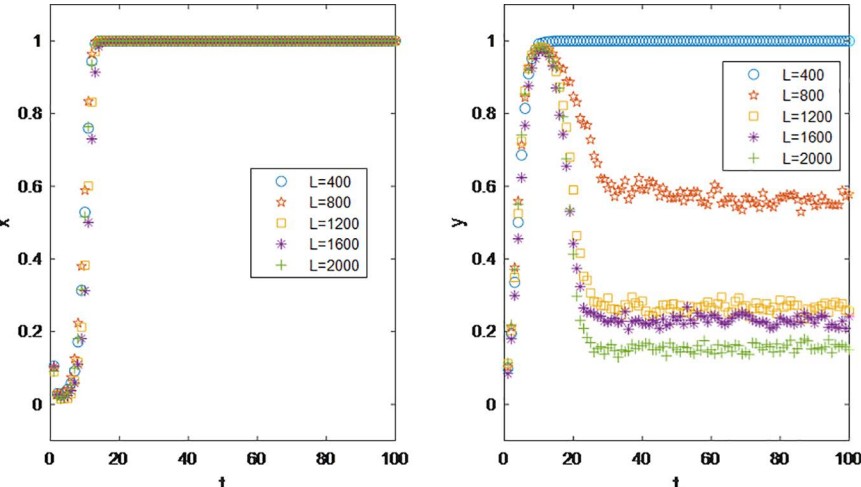

**Fig 10. Impact of Creator incentives _L_ on Parties' Decision Making.**

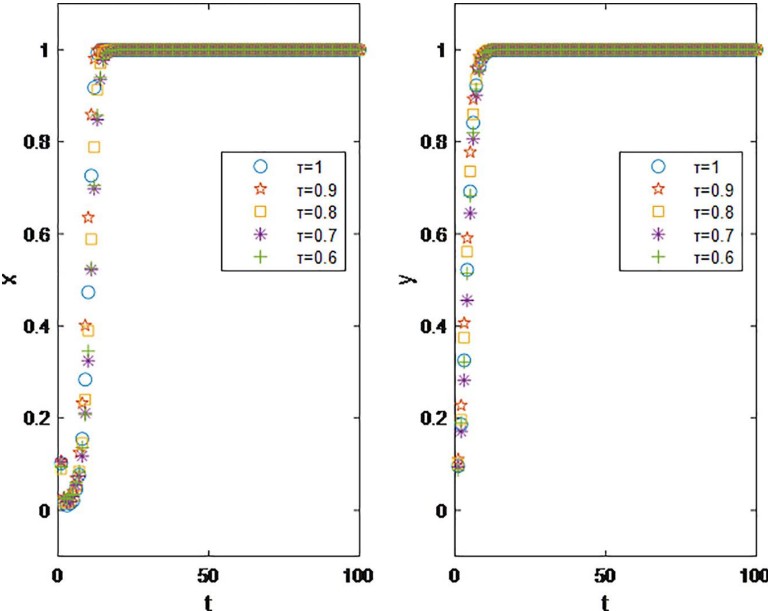

**Fig 11. Impact of Trust level of fans towards brands τ on Parties' Decision Making.**

### 5.3. Example verification

To validate the contribution of our double-layer network model (brand-influencer dynamics) to brand-building, the researchers conducted a case study using the Cosmetics Dataset (https://www.kaggle.com/datasets/kingabzpro/cosmetics-datasets) uploaded by Abid Ali Awan on Kaggle. This empirical analysis demonstrates how the proposed model enhances brand-influencer synergy, providing actionable insights for brand-building strategies.

Due to their broad appeal, high consumption frequency, and accessible price points, beauty products have consistently dominated the online retail sector as a top-selling category. The Cosmetics Dataset (S1 Table) document the price points

of 1,472 cosmetic products across 101 brands, covering four major product types: moisturizer, cleanser, treatment (targeted skincare), and face mask. Leveraging S1 Table, we construct the brand network following these rules:

• **Primary Product Categories & Price Point Calculation:**

For each brand, identify its top two most prevalent product categories (i.e., categories with the highest number of listed products). The average price point (P) of these dominant categories is calculated and assigned as the brand's representative price.

• **Competitive Relationship Definition:**

An undirected edge is established between two brand nodes if they share at least one primary product category, indicating a competitive relationship within overlapping market segments.

The influencer network is constructed by retaining the top 101 nodes by degree from the post-growth tunable clustering scale-free network. Other model parameters—including node link rules, edge-linking probabilities, and clustering coefficients—are configured using empirically validated values from the aforementioned simulation experiments, as detailed in Table 5.

In the final experimental phase, profit trajectories for brands in cooperation, brands not in cooperation, influencers in cooperation, and influencers not in cooperation were plotted across the evolutionary timeline, as shown in Fig 12. The horizontal axis represents the game sequence (t), and the vertical axis indicates normalized profit values. The revenue curves of these four agent categories exhibit significant divergence, reflecting distinct strategic outcomes under the proposed model.

Within the brand cohort, brands in cooperation exhibit a robust growth trajectory from the initial stages, indicating a positive correlation between cooperative strategies and economies of scale. In contrast, the profits of brands not in cooperation persistently hover at the zero-profit boundary, demonstrating the invalidation of defection strategy dominance typically observed in the prisoner's dilemma model.

The influencer cohort exhibits divergent dynamics shaped by strategic cooperation choices. Influencers in cooperation demonstrate profit trajectories characterized by logistic growth, where initial rapid gains reach a local maximum before converging to a suboptimal equilibrium of $1 \times 10^6$ through market-mediated adjustments. This pattern underscores the diminishing marginal returns inherent to cooperative strategies within bilateral markets, as escalating collaboration costs and resource saturation erode incremental benefits. In contrast, influencers not in cooperation sustain profit curves persistently aligned with the zero-value baseline, revealing that creators reliant solely on content production—without brand partnerships—face existential viability challenges in the influencer economy. These contrasting outcomes highlight the critical role of strategic alliances in overcoming market saturation and achieving sustainable monetization, while non-cooperative strategies risk marginalization due to insufficient monetization pathways beyond organic content engagement.

The experimental findings demonstrate that the sustained stability of the influencer economy can indeed generate substantial profits for brands. Notably, influencers' profits peak earlier than those of brands, aligning with real-world dynamics where the core objective of influencer marketing lies in market expansion and identifying potential loyal customers. Initial profit surges are driven by follower-driven effects, which also explains the subsequent decline in influencer earnings: as the system evolves, the influencer market becomes saturated, and brands acquire a stable base of loyal customers through influencer partnerships. Consequently, both parties' profits stabilize, reflecting a mature equilibrium in the influencer-brand ecosystem.

**Table 5. System Simulation Parameter Setting Table.**

| Parameter | x | y | C1 | C2 | Q1 | P | ρ | τ |
|---|---|---|---|---|---|---|---|---|
| initial value | 0.2 | 0.2 | 10 | 1000 | 100 | – | 1 | 0.9 |
| Parameter | λ | α | γ | R | L | δ | K | q |
| initial value | 5% | 0.1 | 10% | 30% | 400 | $\log^{Ni}$ | 0.1 | 0.1 |

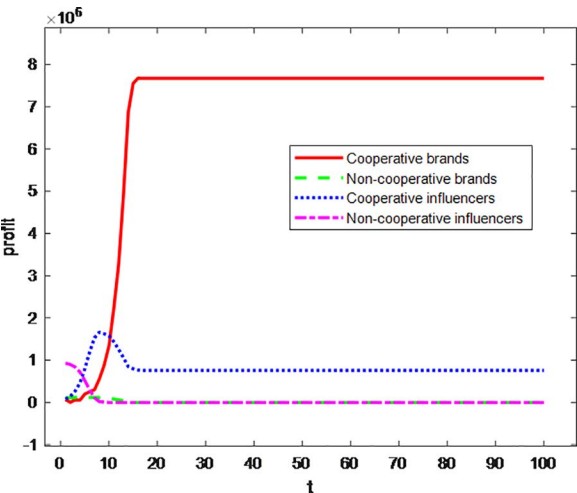

**Fig 12. The profit curves of influencers and cosmetics industry brands.**

## 6. Discussion

The model presented in this paper still has room for optimization. Given that the academic research and discussions on the operational model of influencer marketing are still in their infancy, and constrained by the user privacy policies of online platforms, the construction of the model and the setting of simulation parameters have certain limitations. For the "brand-influencer" double-layer network game model constructed in this paper, the following improvements can be considered in the future. On the one hand, the purchasing power of influencer's followers has not been considered. This study only examined the impact of low-priced fast-moving consumer goods on the cooperative behavior between brands and influencers in the influencer marketing. In reality, there are also brands in the influencer marketing that offer high-priced products such as cars, luxury goods, and travel photography services. One is that the matching degree between the brand and the influencer's labels has not been considered. Apart from general products, each brand's products have clear target customers. For example, the target users of maternal and infant products are families with newborn babies, the target users of e-sports products are more inclined towards males, and the target audience of educational training courses are more inclined towards students. Therefore, it is possible to consider the development trend of the influencer marketing when brands seek influencers that match their product attributes. Third, adopting more flexible and behaviorally consistent rules for the transformation of evolutionary strategies. This paper still uses the traditional method of learning from neighbors' behaviors to update strategies. Future researchers can develop different strategy update rules tailored to the characteristics of the influencer marketing. For instance, whether the brand continues to invest in the next game can be determined based on whether the current game's earnings meet the brand's expected ROI; influencers not only decide whether to accept brand commerce based on income but also consider factors such as their personal affinity with the brand. Finally, the construction method of the model. The double-layer network in this paper is constructed in a "one-to-one" format, while in practice, brands often contact multiple influencers for promotion. Future researchers can further study the cooperation issues between brands and influencers by integrating the above ideas.

## 7. Conclusion

This paper constructs a double-layer complex network game model for the cooperation between brands and influencers, deeply exploring the evolutionary mechanisms of whether brands and influencers cooperate in promoting products within the influencer marketing model. From the micro-node level, it builds the brand network and influencer network, connecting

brands with "consumers" (the influencers' followers) through influencers, and establishes a relational model. It uses the concept of macro-evolutionary game theory to analyze whether influencers and brands collaborate in this model and further studies the impact of relevant variables on cooperation decisions based on simulation experiments. The research findings can be summarized as follows:(1) The influencer marketing is a win-win cooperation model for brands and influencers. Under continuous profit growth and without interference from external factors, both parties quickly evolve from a state of low cooperative intention to full cooperation; (2) Game model parameters influence cooperative decision-making. Brands face a structural disadvantage within the influencer marketing paradigm: while influencer partnerships can yield substantial profits, product-related factors (e.g., C, C2, γ) exert minimal impact on influencers' strategic choices yet significantly shape brands' own decision calculus. Notably, Creator motivation L only affects the decision-making of the master. Therefore, the brand needs to evaluate the relevant indicators with the collaborators while balancing its own factors to make the final cooperation decision. (3) Network topology significantly influences collaborative decision-making between brands and influencers. The structural parameters of complex networks—such as the tunable clustering coefficient (q), which in this study correlates with an influencer's niche specialization and market influence—can be abstracted into real-world relational dynamics (e.g., follower concentration, competitive overlaps). These insights underscore the necessity for both brands and influencers to strategically monitor industry landscapes and maintain adaptive agility, recalibrating partnership strategies in response to environmental shifts such as market saturation or algorithm-driven platform changes;(4) The influencer marketing is an economic model that heavily relies on the number and purchasing power of influencer followers. In some cases, the cooperation curve shows a marginal diminishing effect (e.g., R and L).When system variables are stable, both parties initially have a strong willingness to cooperate, but as follower repurchase rates decline, some individuals shift strategies and choose not to cooperate.

The implications of our research should be considered within the limitations of the study. Although we have made efforts to overcome these limitations, there are still some unavoidable constraints. Based on the comprehensive analysis of this research, to promote the healthy development of the influencer marketing model, increase the benefits for brands within this model, tap into potential users among the influencer's followers, and stimulate market consumption, the following management insights are proposed from the perspective of brands.

First, reasonably control the profit margin of products. Given the numerous stakeholders in the influencer marketing, including not only the brands and influencers discussed in this paper but also content creation platforms, governments, suppliers, online visitors, and third-party agencies (such as MCNs and logistics companies), each party can extract a portion of the brand's benefits at their respective stages. Product commissions are merely a form of profit distribution between brands and influencers in the influencer marketing. Therefore, in practice, brands must control the reasonableness of product pricing and consider the costs of all parties to achieve their own profitability.

Second, rationally allocate additional investment in influencer promotion to reduce costs and increase efficiency. The influencer marketing is a promotion model characterized by low investment, high exposure, and short cycles. By collaborating with influencers, brands can gain more exposure opportunities, increase user stickiness, and enhance user loyalty, thereby achieving better marketing effects. However, the potential value of influencer followers is limited, and excessive investment in the short term will not yield higher conversions. Therefore, brands should formulate appropriate scales for promotion expenses, media recruitment, quality control, etc., suitable for their brand size.

Third, brands should maintain stable and long-term cooperative relationships with influencers with high follower counts. For easily saleable products with low unit prices, influencers with high follower counts have more stable sales performance. Establishing good cooperative relationships helps both parties continuously gain benefits in the influencer marketing. At the same time, brands should clearly define common marketing goals and expected results with influencers. This helps ensure that both parties maintain consistency and focus during the cooperation process; respect the professional knowledge and skills of influencers, giving them full trust and autonomy. Avoid excessive interference in their creative process, allowing them to freely showcase their unique style and charm. Regularly evaluate and provide feedback on

cooperation effects, understanding the strengths and weaknesses in the cooperation process. Work together with influencers to develop improvement plans, continuously optimize cooperation strategies and content, and enhance cooperation effectiveness. Finally, brands should clearly define the contract labor billing methods with cooperative influencers. This paper adopts the currently convenient and quantifiable method of settlement based on the percentage of followers, as well as methods based on influencer influence and interaction levels, to ensure cooperation on a mutually satisfactory basis.

## Supporting information

**S1 Table. The cosmetics dataset used for example validation.**
(CSV)

**S2 Table. The values used to build fig 3–12.**
(XLSX)

## Author contributions

**Conceptualization:** Zeguo Qiu.

**Funding acquisition:** Zeguo Qiu.

**Investigation:** Yunhao chen, Hao Han, Tianyu Wang.

**Methodology:** Yunhao chen, Hao Han, Tianyu Wang.

**Project administration:** Zeguo Qiu.

**Software:** Yunhao chen, Hao Han.

**Visualization:** Yunhao chen, Hao Han, Tianyu Wang.

**Writing – original draft:** Yunhao chen, Hao Han, Tianyu Wang.

**Writing – review & editing:** Zeguo Qiu.

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
