## [Decision Letter · Decision Letter 0]

17 Jan 2025

Dear Dr. Qiu,

Thank you for submitting your manuscript to PLOS ONE. After careful consideration, we feel that it has merit but does not fully meet PLOS ONE’s publication criteria as it currently stands. Therefore, we invite you to submit a revised version of the manuscript that addresses the points raised during the review process.

We look forward to receiving your revised manuscript.

Kind regards,

Tyler Horan

Academic Editor

PLOS ONE

Journal Requirements:

2. Thank you for stating the following financial disclosure: This research was supported in part by the Philosophy and Social Planning Project of Heilongjiang Province under Grant Nos. 22GLB105 and 23GLA011.

Additional Editor Comments:

Dear Zeguo Qiu,

After consideration by a a number of reviewers, I have elected to require major revisions to this article. While the paper offers valuable insights into influencer marketing using a novel theoretical framework, it requires substantial revision to improve clarity, rigor, and applicability. Addressing the issues below and in the reviewer comments will strengthen its contribution to both academic literature and practical applications.

1. The language in several sections is overly technical, which might hinder comprehension for readers outside the field.

2. The literature review is heavily skewed towards industry reports and lacks sufficient academic citations. Furthermore, key theoretical foundations like influencer marketing or network game theory are underdeveloped.

3. The reliance on simulations without validation against real-world data limits the practical credibility of the findings.

As for the writing, there are redundancies and inconsistencies, particularly in the introduction and methodology sections. Before further submission, please ensure that the reviewers comments are addressed.

Best Regards,

Tyler Horan

Reviewers' comments:

Reviewer's Responses to Questions

**Comments to the Author**

1. Is the manuscript technically sound, and do the data support the conclusions?

Reviewer #1: Yes

Reviewer #2: No

2. Has the statistical analysis been performed appropriately and rigorously?

Reviewer #1: Yes

Reviewer #2: No

3. Have the authors made all data underlying the findings in their manuscript fully available?

Reviewer #1: Yes

Reviewer #2: No

4. Is the manuscript presented in an intelligible fashion and written in standard English?

Reviewer #1: Yes

Reviewer #2: No

Reviewer #1: Both the title (‘Research on Influencer Marketing Strategies Based on Double-Layer Network Game Theory’) and the abstract are well structured with an adequate identification of what is subsequently addressed in the text.

It is an article that contributes to society and helps to promote knowledge about the importance and role of influencers in the construction and tailoring of marketing strategies and, therefore, of their orientation towards society and consumers.

The objectives and hypotheses are well stated and appropriate, with methodological rigour and well-designed and thought-out research instruments.

The results are clearly presented, and it is a solid article, with a good presentation of the objectives. It is therefore a clearly publishable article. In any case, I suggest that the following texts be considered and reviewed:

- Gurrieri, L., Drenten, J., & Abidin, C. (Eds.). (2024). Influencer Marketing: Interdisciplinary and Socio-Cultural Perspectives. Taylor & Francis.

- Jiménez-Marín, G., Sanz-Marcos, P., & Tobar-Pesantez, L. B. (2021). Keller's resonance model in the context of fashion branding: persuasive impact through the figure of the influencer. Academy of Strategic Management Journal, 20 (6).

- Lee, J., Walter, N., Hayes, J. L., & Golan, G. J. (2024). Do Influencers Influence? A Meta-Analytic Comparison of Celebrities and Social Media Influencers Effects. Social Media+ Society, 10(3), 20563051241269269.

- Sanz Marcos, P., Jiménez-Marín, G., & Elías-Zambrano, R. E. (2021). Aplicación y uso del Modelo de Resonancia o Customer-Based Brand Equity (CBBE). Estudio de la lealtad de marca a través de la figura del influencer. methaodos. revista de ciencias sociales, 9(2), 200-218.

- Zozaya-Durazo, L., Feijoo, B., & Sádaba, C. (2023). The Role that Influencers Play in Consumption Decisions Made by Minors. Doxa Comunicación, 36, 401-413.

Reviewer #2: The current version is not acceptable for publication in PLOS ONE. This current version does not covey any novel findings. At the same time the figures are not well-prepared. Need to improve all the sections carefully with scientific insights.

**Do you want your identity to be public for this peer review?** For information about this choice, including consent withdrawal, please see our Privacy Policy

Reviewer #1: No

Reviewer #2: No

---

## [Author Response · Author response to Decision Letter 1]

31 Mar 2025

Dear editor Tyler Horan.

On behalf of my co-authors, we thank you very much for giving us an opportunity to revise our manuscript� we appreciate editor and reviewers very much for their positive and constructive comments and suggestions on our manuscript entitled “Research on Influencer Marketing Strategies Based on Double-Layer Network Game Theory”.(Submission ID PONE-D-24-46619)

We have studied the Reviewer's comments carefully and have made revision which marked in tracked changes in the revised submission file. We have tried our best to revise our manuscript according to the comments. We would like to submit for your kind consideration.

Thank you and best regards.

Yours sincerely.

Corresponding author.

Name: Zeguo Qiu

E-mail: qiuzg@hrbcu.edu.cn

---

## [Decision Letter · Decision Letter 1]

28 May 2025

Research on Influencer Marketing Strategies Based on Double-Layer Network Game Theory

PONE-D-24-46619R1

Dear Dr. Qiu,

We’re pleased to inform you that your manuscript has been judged scientifically suitable for publication and will be formally accepted for publication once it meets all outstanding technical requirements.

Kind regards,

Tyler Horan

Academic Editor

PLOS ONE

Additional Editor Comments (optional):

Reviewers' comments:

Reviewer's Responses to Questions

**Comments to the Author**

Reviewer #1: All comments have been addressed

Reviewer #3: (No Response)

2. Is the manuscript technically sound, and do the data support the conclusions?

Reviewer #1: Yes

Reviewer #3: Yes

3. Has the statistical analysis been performed appropriately and rigorously?

Reviewer #1: Yes

Reviewer #3: Yes

4. Have the authors made all data underlying the findings in their manuscript fully available?

Reviewer #1: Yes

Reviewer #3: Yes

5. Is the manuscript presented in an intelligible fashion and written in standard English?

Reviewer #1: Yes

Reviewer #3: Yes

Reviewer #1: All the considerations that were raised with regard to authorship correspond, in the current text, to a revised version. Check. And in this sense, for my part, the article is suitable for publication.

Reviewer #3: Thank you for the opportunity to read such an interesting paper related to Influencer Marketing Strategies Based on Double-Layer Network Game Theory. The paper is very interesting and addresses a novel topic. However I would have some suggestions to improve it. In my opinion the abstract is too long and I did not clearly find the main objectives to this study. The final part of the paper should also include, in my opinion, discussion of the results obtain though the lens of other research in the field (links with other research with results similar or opposite). At the same time, the scientific novelty of the paper is not very clear, nor are the implication for literature in this field.

**Do you want your identity to be public for this peer review?** For information about this choice, including consent withdrawal, please see our Privacy Policy

Reviewer #1: No

Reviewer #3: No

---

## [Editor Report · Acceptance letter]

PONE-D-24-46619R1

PLOS ONE

Dear Dr. Qiu,

I'm pleased to inform you that your manuscript has been deemed suitable for publication in PLOS ONE. Congratulations! Your manuscript is now being handed over to our production team.

Kind regards,

on behalf of

Dr. Tyler Horan

Academic Editor

PLOS ONE